# CITK Loss Inhibits Growth of Group 3 and Group 4 Medulloblastoma Cells and Sensitizes Them to DNA-Damaging Agents

**DOI:** 10.3390/cancers12030542

**Published:** 2020-02-26

**Authors:** Gianmarco Pallavicini, Giorgia Iegiani, Gaia Elena Berto, Elisa Calamia, Edoardo Trevisiol, Andrea Veltri, Simona Allis, Ferdinando Di Cunto

**Affiliations:** 1Neuroscience Institute Cavalieri Ottolenghi, 10043 Turin, Italy; 2Department of Neurosciences, University of Turin, 10126 Turin, Italy; 3Neuroscience Institute of Turin (NIT), 10043 Turin, Italy; 4Department of Radiology, San Luigi Gonzaga Hospital, University of Turin, 10043 Turin, Italy; 5Radiation Oncology, Department of Oncology, San Luigi Gonzaga Hospital, University of Turin, 10043 Turin, Italy

**Keywords:** medulloblastoma, radiation, cisplatin, TP53, 53BP1, genomic instability, double strand break, DBS, HR

## Abstract

Medulloblastoma (MB) is the most common malignant brain tumor in children, and it is classified into four biological subgroups: WNT, Sonic Hedgehog (SHH), Group 3 and Group 4. The current treatment is surgery, followed by irradiation and chemotherapy. Unfortunately, these therapies are only partially effective. Citron kinase protein (CITK) has been proposed as a promising target for SHH MB, whose inactivation leads to DNA damage and apoptosis. D283 and D341 cell lines (Group 3/Group 4 MB) were silenced with established siRNA sequences against CITK, to assess the direct effects of its loss. Next, D283, D341, ONS-76 and DAOY cells were treated with ionizing radiation (IR) or cisplatin in combination with CITK knockdown. CITK depletion impaired proliferation and induced cytokinesis failure and apoptosis of G3/G4 MB cell lines. Furthermore, CITK knockdown produced an accumulation of DNA damage, with reduced RAD51 nuclear levels. Association of IR or cisplatin with CITK depletion strongly impaired the growth potential of all tested MB cells. These results indicate that CITK inactivation could prevent the expansion of G3/G4 MB and increase their sensitivity to DNA-damaging agents, by impairing homologous recombination. We suggest that CITK inhibition could be broadly associated with IR and adjuvant therapy in MB treatment.

## 1. Introduction

High-grade brain tumors (HGBT) represent an important unmet medical challenge. In pediatric age, the most common HGBT is medulloblastoma (MB), which has been classified into four biological subgroups, based on microarray and genomic sequencing technologies (WNT, SHH, Group 3 and Group 4) [1,2,3]. WNT and SHH MB subgroups are primarily driven by mutations leading to constitutive activation of the Wingless and Sonic Hedgehog signaling pathways, respectively. WNT and SHH MBs are clearly separable across the majority of transcriptional and methylation profiling studies, demonstrating minimal overlap with other subgroups [1]. The transcriptomes of Group 3 and Group 4 medulloblastoma are more similar to each other since several cytogenetic features, such as isochromosome 17q (i17q), are found in both groups [2]. However, the genetics and biology underlying Group 3 and Group 4 MB remain less clear.

MB is currently treated with surgery, followed by irradiation of the entire neuroaxis and high-dose multi-agent chemotherapy. Long-term survival rates can be as high as 90% in the rare WNT subgroup, but they are usually around 50% in most other cases, with an intermediate prognosis in Group 4 and worse in Group 3 patients [3]. Thus, many patients still die despite these treatments and those who survive suffer from neurological, cognitive and endocrine disorders caused by the aggressive therapy [3].

For these reasons, more effective and specific therapies are urgently needed. A possible strategy to develop new anticancer therapies is to directly target the molecular pathways altered by driver mutations. Targeted therapy has been developed for the SHH MB, which occurs in 25% of cases and is the better understood subtype [4]. However, only a subgroup of these patients responds to small-molecule inhibitors of the SHH pathway, and even in these cases, resistance rapidly develops [5,6]. No targeted therapies have so far been identified for Group 3 and 4 cases.

An alternative strategy for drug development could be to target molecules that, despite not being mutated, are nevertheless required for tumor growth and progression [7]. Cancer cells may be specifically dependent on non-mutant proteins responsible for specific biological features of the starting normal cells, or for biological changes induced by the oncogenic process [7]. Although the cellular origin of the different MB subtypes is still debated, it is clear that MB cells share many molecular features with cerebellar granule progenitors and radial glia cells [8,9]. Genes mutated in primary microcephaly syndromes (MCPH) are attractive targets for brain-tumor-directed drug development [10,11]. MCPH is a rare and genetically heterogeneous disorder, in which brain volume is strongly reduced, as compared to the rest of the body [12]. Studies performed in the last two decades have led to the identification of 20 MCPH genes [13]. A striking common feature of these genes is that they are selectively required for proliferation and genomic stability of neural progenitors, despite being expressed in all proliferating cell types [14]. In all cases, cortical radial glia cells are severely and specifically affected by the gene defect, as compared to most other cells of the developing fetus. The biological basis of this specificity is only partially understood. In many cases, MCPH proteins are associated with centrosomes, and their loss leads to cell-cycle and mitosis delay, mitotic failure or randomization of spindle orientation [15]. It turn, these alterations may tilt the balance between symmetric and asymmetric divisions of neural stem cells, decreasing the pool of proliferating neural progenitors and/or increasing the frequency of premature commitment or terminal differentiation [15]. It has also been shown that loss of MCPH proteins leads to accumulation of DNA damage and apoptosis [16,17,18]. A subset of microcephaly syndromes shows strong cerebellar hypoplasia, indicating that cerebellar progenitors are sensitive to inactivation of the particular genes. These include ZNF335 (MCPH10 [19,20]), ANKLE2 (MCPH16 [21]), MFSD2A (MCPH15 [22]), CENPE (MCPH13 [23]), KIF14 (MCPH20 [24]) and CIT (MCPH17 [25,26]). The latter gene could be particularly interesting, since its main product (Citron kinase, CITK) is a ser/thr kinase, whose activity is essential for function [25,26].

CITK is required in neural progenitors for cytokinesis [27,28,29] and mitotic spindle positioning [30]. In addition, CITK loss leads to accumulation of DNA double strand breaks (DSB), strongly activating the P53 signaling pathway in developing central nervous system [16]. We have shown that CITK loss induces apoptosis and cell senescence in SHH MB cells and reduces growth of both xenograft and transgenic MB [31]. Interestingly, these anti-proliferative effects of CITK loss may be engaged through TP53-dependent and TP53-independent mechanisms [16,31].

In this report, we analyzed the effects of CITK deletion in Group 3/4 MB cell lines. Our results support CITK as a drug target for all types of MB, as it emerges to have a significant effect even on these cells. Moreover, we show that the reduction of CITK expression impairs homologous recombination and potentiates the effects of IR or cisplatin in reducing growth potential and colony forming activity of MB cells. These results suggest that CITK inactivation may increase the effectiveness of established treatments for MB.

## 2. Results

### 2.1. CITK Knockdown Impairs Proliferation and Cytokinesis in Group 3 and Group 4 MB

To investigate whether CITK could be a potential target for Group 3 and Group 4 (G3/G4) medulloblastoma (MB), we resorted to the established human medulloblastoma cell lines D283 (*p53 wt, no myc amplification*) and D341 (*p53 wt, myc amplification*) [32]. Transient transfection of validated siRNAs (siCITK1 and siCITK2) [31] was effective in downregulating CITK protein levels in both cell lines, as assessed by Western blot, after 72 h (Figure 1A). In the D283 line, cell expansion was significantly reduced by CITK downregulation after 100, 150 and 200 h from transfection (Figure 1B). A similar effect was observed in D341 cells, although in this case, the kinetic of proliferative impairment was faster than in the D283 line (Figure 1C). Western blot analysis revealed that CITK levels were still suppressed by both siRNAs after 200 h for D283 and 144 h for D341 (Appendix A). Since cytokinesis failure is the most knowneffect of CITK loss in both sensitive normal tissues and tumor cells [28,33], we investigated whether this phenotype also occurs in G3/G4 MB cell lines. As shown in Figure 1D,G, CITK downregulation significantly increased the percentage of D283 and D341 cells with two nuclei, 100 and 72 h after transfection, respectively. On average, the integrated DAPI intensity in binucleated cells was double than in mononucleated cells (Appendix A), highlighting a failure in mitosis after anaphase.

### 2.2. CITK Knockdown Increases Apoptosis and Cell-Cycle Arrest in D283 and D341 Cell Lines

Besides cytokinesis failure, it has previously been reported that CITK loss leads to apoptosis via TP53 [16,31,33]. We therefore tested whether these events are also induced in G3/G4 cell lines. Both D283 and D341 CITK-depleted cells showed increased levels of phospho-TP53, as well as of the TP53 targets BAX and P21 (Figure 2A–C and Appendix A). Flow-cytometry analysis revealed that, in both lines, CITK depletion leads to a significant increase of cells in the G0/G1 phase and a reduction of the G2/M peak (Figure 2D,E), indicating a strong inhibition of cell-cycle progression. The proliferative block may justify a reduction of the G2/M in spite of an increased proportion of binucleated cells, as detected by IF (Figure 1D–G). Ultimately, D283 and D341 showed increased levels of apoptosis, as indicated by increased percentage of cells positive for TUNEL (Figure 2F,G) and cleaved Caspase 3 (Appendix A and [34]), as well as by increased total levels of cleaved Caspase 3 (Figure 2A–C).

### 2.3. CITK Loss Induces Double Strand Breaks Accumulation in MB Cells

In previous studies, we showed that CITK knockdown primarily leads to accumulation of DNA double strand breaks (DSB) in neural progenitors and HeLa cells, both in vivo and in vitro [16,31]. We therefore asked whether markers of DNA damage are also altered in G3/G4 MB cells. CITK knockdown induced a significant increase of γH2AX levels (Figure 3A–C), as well as of γH2AX nuclear foci (Figure 3D,E), which are indicative of increased DNA damage. Interestingly, these cells exhibit a parallel decrease of the DNA repair protein RAD51 (Figure 3A,C). In addition, we observed a significant increase in the frequency of nuclear foci positive for 53BP1 (Figure 3F,G), indicating that many of the DNA lesions induced by CITK loss were DSBs [35,36]. Accumulation of 53BP1 foci 48 h after CITK knockdown was also observed in ONS-76 and DAOY cells (Appendix A), in which γH2AX activation was previously demonstrated [31]. The latter result indicates that DSBs accumulation is a common consequence of CITK loss in MB cells, which may be followed by apoptosis or cell-cycle arrest.

### 2.4. CITK Knockdown Strongly Reduces Nuclear RAD51 Levels in MB Cells and Impairs Homologous Recombination

RAD51 is a crucial player in homologous recombination (HR)-dependent DSB repair [37]. The finding of reduced total levels of this protein suggests that DSB accumulation detected in MB cells could be caused by reduced efficiency of HR-dependent repair pathway. Since RAD51 operates in the nuclear compartment and its loss induces DNA damage and radiosensitization [38], we set out to evaluate nuclear RAD51 levels in CITK-depleted MB cells. To this aim, we resorted to ONS-76 and DAOY, which we previously engineered for conditionally expressing CITK-specific shRNAs [31]. In these cells, profound CITK depletion can be induced and maintained more efficiently than after transient transfection of siRNAs (Appendix A), thus simplifying the cell fractionation protocol. Even in this case, we found that RAD51 total levels are reduced after CITK loss, although to a lesser extent if compared with D283 and D341 cells (Appendix A). Nevertheless, in both cell lines, nuclear RAD51 were strongly reduced (Figure 4A,B). In particular, the reduction was around 60% for ONS-76 shCITK and 50% for DAOY shCITK (Figure 4B,D). To consolidate this finding on G3/G4 MB cell lines, we evaluated the frequency of nuclear RAD51 accumulations by immunofluorescence analysis, which was significantly reduced in both cell types (Figure 4C,D).

To directly evaluate whether HR activity is impaired by CITK loss, we resorted to a functional HR assay [39,40,41]. HR efficiency was assessed by semiquantitative PCR, after co-transfection of two plasmids (dl-1 and dl-2) possessing homologous sequences. CITK knockdown significantly reduced the formation of the HR product, if compared to control cells (Figure 4E,F). This result strongly suggests that CITK prevents genomic instability through HR-mediated DNA repair.

### 2.5. CITK Downregulation Potentiates the Effects of Ionizing Radiation and Cisplatin in Inhibiting MB Cell Growth

A crucial point to consolidate CITK as a useful target for therapy is to investigate whether its inactivation may increase the effectiveness of established treatments. Since CITK knockdown leads to accumulation of DSB and interferes with HR-dependent DNA repair, we investigated the effects of combining CITK depletion with other treatments that kill tumor cells by increasing DSB load. In particular, we tested ionizing radiations (IR), which act through ROS production and are the most effective current treatment [42,43,44]. Moreover, we assessed the effect of combining CITK inactivation with administration of cisplatin, which induces DSB without increasing ROS production and is used for adjuvant chemotherapy in high risk patients [42,43,44].

For all MB lines, we tested different IR doses, to assess the best window for detecting combined effects in clonogenic assays. As reported in the literature, D283 and D341 are more sensitive to radiation treatment compared to ONS-76 and DAOY [45,46,47]. We selected 2 and 4 Grays (Gy) dosage for the first two cell lines, because no colonies were detectable at higher doses in CITK-depleted cells. Based on similar considerations, 4, 6 and 8 Gy were used for ONS-76 and DAOY cells. Likewise, we set out to treat all MB cells only with 1 and 10 µM of cisplatin for 1 hour, since not many colonies were formed at higher concentrations.

After IR or cisplatin treatment, 6,000 cells were plated in six-well plates for D283 and 10,000 cells for D341, to obtain 500 and 300 colonies, respectively (plating efficiency: 5%). For ONS-76 and DAOY, 500 cells were plated to obtain 300 colonies in control cells not irradiated or treated (plating efficiency: 50%). The depletion of CITK, without additional treatment, reduced colony formation capacity by 62% in D283, 25% in D341, 57% in ONS-76 and 30% in DAOY (Appendix A). To highlight the possible combined effects of CITK knockdown with IR or cisplatin treatments, we compared the reduction of colony-forming efficiency induced by treatments in absence or presence of anti-CITK siRNA (Figure 5 and Figure 6). Cells transfected with control and anti-CITK siRNAs without treatments were set as reference for the same transfections combined with IR or cisplatin treatments. Therefore, the strength of the associated effects is represented by the increased slope of the combined treatment curve (siCITK and DNA-damaging agents), as compared to the single treatment curve (siCtrl and DNA-damaging agents). In all the tested cell lines, CITK loss significantly strengthened the reduction of clonogenic potential induced by IR and cisplatin, with shrinkage of both colonies’ number and colonies’ size (Figure 5 and Figure 6). Interestingly, the combined effects were more pronounced in ONS-76 and DAOY cell lines (Figure 6), especially when low levels of CITK were stably maintained, even after the administration of treatments (Appendix A).

Altogether, these data indicate that the combination of IR or cisplatin with CITK inactivation potently reduces the clonogenic growth rate and clonogenic potential of MB cells.

## 3. Discussion

In our previous work, we obtained in vitro and in vivo proof of concept about the suitability of CITK as a potential candidate for SHH MB treatment [31]. Since Group 3 and 4 MB are more frequent than SHH MB and have worse prognosis, we investigated whether the spectrum of cases that could benefit from CITK inhibition may extend beyond SHH MB. The data obtained in human D283 and D341 cells confirmed that CITK is necessary for their in vitro expansion and that CITK loss leads to high frequency of cytokinesis failure and apoptosis, similarly to what has previously been described in developing neural progenitors [16,26,29,48] and in SHH MB [31]. We also confirmed that DNA damage and TP53 activation are prominent consequences of CITK loss, since accumulation of γH2AX and 53BP1 nuclear foci were detected in D283 and D341 cells, besides to ONS-76 and DAOY cells.

The mechanisms leading to these phenotypes remain to be fully determined. In our previous work, we showed that DSBs accumulation in proliferating cells sensitive to CITK loss correlates with reduced RAD51 recruitment at normally recessed DNA ends [16,27]. The present data confirm and extend those findings. We here showed that RAD51 levels were consistently reduced in all the tested MB cell lines. Moreover, we found that nuclear RAD51 levels are reduced by CITK loss even more severely than cytoplasmic levels. Furthermore, less RAD51 foci were present after CITK knockdown, despite an increased number of DSB. The effect of CITK knockdown on RAD51 pointed to the potential role of CITK in HR-dependent DNA repair. This possibility was directly explored by using a functional HR assay, which confirmed a significant reduction of HR efficiency in CITK-depleted MB cells (Figure 4E,F). The latter result would also explain why CITK knockdown is by itself capable of inducing genomic instability [16,31]. CITK may directly impair RAD51 nuclear-cytoplasmic shuttling or nuclear RAD51 stability [38,49,50,51]. These mechanisms would be consistent with the fact that CITK is capable of forming a physical complex with RAD51, especially when CITK catalytic activity is compromised [16]. We cannot exclude that CITK may also affect RAD51 expression at the transcriptional level. It is also possible that CITK may affect other proteins involved in HR. Many different mitotic partners of CITK have been so far described [27,28], while the proteins that interact with nuclear CITK during S and G2 phases are still largely unexplored. Moreover, considering the cell-cycle block which we detected in CITK knockdown cells, it is also possible that the reduced levels of RAD51 and HR efficiency are a consequence of DSB-dependent cell-cycle arrest, rather than a cause DSB accumulation. Indeed, it is well-known that RAD51 expression and HR efficiency are highest in cycling cells [52]. Although more studies are clearly necessary to unravel the mechanistic relationships between these phenomena, our results further underscore the central role of DNA damage and repair in the biological role of CITK in dividing cells.

A second crucial point to propose CITK inactivation as an adjuvant strategy for MB treatment was to assess whether it may increase the effectiveness of established anti-MB treatments, such as IR and DNA cross-linking agents, to increase MB cells’ death. Importantly, in all the MB cell models which we tested, the reductions of clonogenic capability and cell growth determined by IR and cisplatin were significantly more pronounced after CITK knockdown than after treatment with control siRNA, at a greater extent than it would be expected in the case of an additive effect. The effect was highest in ONS-76 and DAOY cell lines constantly depleted of CITK, and also after treatment with DNA-damaging agents, suggesting that CITK not only potentiates the acute consequences of treatments, but is also important in the recovery phase. This observation should be taken into consideration in devising the most appropriate treatment schemes, once CITK-specific inhibitors become available.

In this regard, despite the fact that it was implicated in cell division since 1998, CITK has long been overlooked as a possible target for cancer therapy [33,53,54,55,56,57]. The main reasons for this neglect are probably the highly tissue-specific developmental function, the association with a severe developmental syndrome [48] and the absence of tumor-promoting mutations [33]. Recent data, obtained by our group and other investigators, are likely to change the general interest about this protein. We have shown that, in the case of MB, the tissue-specificity of CITK requirement could be an advantage, since temporally controlled genetic deletion resulted in measurable antitumor activity without evidence of adverse effects [31]. Other studies found that CITK loss can exert anti-neoplastic effects, even in cells whose normal counterparts do not require it [33,53,55,56,57]. Analysis of the COSMIC database revealed that CITK is significantly under-mutated in cancer [33], confirming that it is not a driver, but also indicating that it may play an important tumor-permissive role. Genetic evidence [26], as well as rescue experiments in MB cells [31], suggest that kinase activity is essential for physiological and tumor-sustaining function. Taken together, these studies put forward that it could be worth evaluating CITK inhibitors for their effectiveness in MB and other tumor types. The development of such molecules will therefore represent a crucial topic for future research.

## 4. Materials and Methods

### 4.1. Cell Culture

D341 cells were obtained from ATCC and were cultured in MEM medium (Euroclone, Milan, Italy) supplemented with 20% FBS (Gibco, Gaithersburg, MD, USA), nonessential amino acids (Gibco), L-glutamine (Gibco), sodium pyruvate (Gibco) and 1% penicillin/streptomycin (Gibco). D283 cell line and DAOY were obtained from ATCC and were cultured in MEM medium (Euroclone) supplemented with 10% FBS (Gibco), nonessential amino acids (Gibco), L-glutamine (Gibco), sodium pyruvate (Gibco) and 1% penicillin/streptomycin (Gibco). ONS-76 cells were cultured in RPMI medium (Euroclone) supplemented with 10% FBS (Gibco) and 1% penicillin/streptomycin (Gibco). ONS-76 and DAOY conditionally expressing control and CITK specific shRNAs were previously generated by Pallavicini et al. [31]. All cells were grown at 37 °C, in a humidified incubator, with 5% CO_2_.

### 4.2. Transfection and RNAi

In order to silence CITK expression, previously published CITK double-stranded RNAs (siCITK1 and siCITK2) and nontargeting siRNA were used (Dharmacon, Lafayette, CO, USA) [31,58]. Cells were placed on six-well plates and were transfected, using 6.25 µL of the required siRNA (20 μmol/L), together with 1.5 µL of Lipofectamine 2000 (Invitrogen, Carlsbad, CA, USA) according to the manufacturer’s instructions. Efficient knockdown was obtained after 48 h for ONS-76 and DAOY and after 72h for D283 and D341 cell lines.

### 4.3. Analysis of Cell Proliferation

To analyze cell proliferation, 1 × 10^4^ D283 cells were seeded into 24-well plates, transfected with siCtrl, siCITK1 and siCITK2 and counted in triplicates, after 100, 150 and 250 h.

Then, 8 × 10^4^ D341 cells were seeded into 6-well plates, transfected with siCtrl, siCITK1 and siCITK2 and counted, after 72 and 144 h, in triplicates, with Biorad TC20 Automated Cell Counter (Bio-Rad, Hercules, CA, USA).

### 4.4. Antibodies

The following antibodies were used: mouse monoclonal anti-citron (#611377; Transduction Laboratories, BD Biosciences, Franklin Lakes, NJ, USA), mouse monoclonal anti-vinculin (#V9131; Sigma-Aldrich, Darmstadt, Germany), mouse mono- clonal anti-a-tubulin (#T5168; Sigma-Aldrich), rabbit monoclonal anti-53BP1 (#ab36823, Abcam, Cambridge, MA, USA), rabbit monoclonal anti-RAD51 (#sc-8349, Santa Cruz, Dallas, TX, USA), rabbit polyclonal anti-γH2AX (S139; 20E3; #2577; Cell Signaling Technology, Danvers, DA, USA), mouse monoclonal anti-P21 (#sc-6246, Santa Cruz), rabbit polyclonal anti-cleaved caspase-3 (#9661S; Cell Signaling Technology,), mouse monoclonal anti-TP53 ( Clone 1C12; #2524S, Cell Signaling), rabbit monoclonal anti pTP53 (#9284S; Cell Signaling Technology), rabbit monoclonal anti-BAX (#sc-526, Santa Cruz) and mouse monoclonal Lamin A/C (Clone 4C11; SAB4200236, Sigma-Aldrich).

### 4.5. Western Blotting

To obtain a whole cell lysate, cells were lysed in RIPA buffer (1% NP40, 150 mmol/L of NaCl, 50 mmol/L of Tris-HCl pH 8, 5 mmol/L of EDTA, 0.01% SDS, 0.005% sodium deoxycholate, Roche protease inhibitors and PMSF) for 10 minutes, at 4 °C. To separate nuclei from the cytoplasm, the Nuclear Fractionation protocol from Abcam was used (https://www.abcam.com/ps/pdf/protocols/Nuclear%20fractionation%20protocol.pdf). Briefly, cells were lysed in Buffer A (10 mM of HEPES, 1.5 mM of MgCl_2_, 10 mM of KCl, 0.5 mM of DTT and 0.05% NP40 (or 0.05% Igepal or Tergitol), pH 9.7) for 10 min, on ice. Then cells were centrifuged at 3000 rpm for 10 min, at 4 °C, and the supernatant was aliquoted as the cytoplasmatic fraction. Pellet was resuspended with Buffer B (5 mM of HEPES, 1.5 mM of MgCl_2_, 0.2 mM of EDTA, 0.5 mM of DTT, 26% Glycerol (v/v), pH 7.9) and 4.6 M of NaCl. For immunoblots, equal amounts of proteins from both whole-cell lysates and nucleus-cytoplasm fraction were resolved by SDS–PAGE and blotted to nitrocellulose membranes.

### 4.6. Immunofluorescence

Then, 50,000 D341 treated cells resuspended in PBS were spun in Thermo Scientific Cytospin 4 (Thermo Fisher Scientific, Waltham, MA, USA) at 500 rpm, for 10 min, and then processed for immunofluorescence. All cell-lines cells were either fixed for 10 min at RT with PFA 4% or, for γH2AX, 53BP1 and RAD51 staining, fixed 5 minutes at RT, using PFA 2%, treated for 10 minutes at RT using CSK buffer (100 mmol/L of NaCl, 300 mmol/L of sucrose, 3 mmol/L of MgCl2, 10 mmol/L of PIPES (pH 6.8) and 0.7% Triton), and fixed again, for 5 minutes, at RT, using PFA 2%. Subsequently, cells were permeabilized with 0.1% Triton X-100, in PBS, for 10 min, saturated in 5% BSA in PBS for 30 min, and incubated with a primary antibody for 2 h, at RT. Primary antibodies were detected with anti-rabbit Alexa Fluor 488 or 555 (Molecular Probes, Invitrogen), anti-mouse Alexa Fluor 488 or 555 (Molecular Probes, Invitrogen), used at 1:1000 dilution, for 30 min. Cells were counterstained with 0.5 mg/mL of DAPI for 10 min and washed with PBS. Finally, cell slides were mounted with Prolong (Thermo Fisher Scientific). TUNEL assay was performed 72 h after transfection in D283 and D341, using the TMR red In Situ Cell Death Detection Kit (Roche, Basel, Switzerland), according to the manufacturer’s protocol.

### 4.7. Cell-Cycle Analysis

D283 cells and D341 cells were plated in 6-well plates and transfected with siCtrl and siCITK 1 for 100 h and 72 h respectively. They were then fixed in cold 70% ethanol overnight and then washed with PBS two times and stained with 5 µL of PI (from 1 mg/mL of stock) in 500 µL of PBS and RNAse.

### 4.8. Homologous Recombination Assay

To analyze homologous recombination after CITK knockdown, a Homologous Recombination Assay Kit (Cat. 35600, Norgen Biotek Corporation, Thorold, ON, Canada) was used, following manufacturer instructions. Briefly, 100,000 D283 cells and 50,000 D341 cells were plated in 24-well plates and transfected with siCtrl, siCITK 1 and siCITK2 for 72 h. Then, 50,000 ONS-76 and DAOY cells were instead plated in 24-well plates and transfected with siCtrl and siCITK1 for 48 h. After CITK knockdown, cells were transfected with 0.5 ug of dl plasmids (dl-1 and dl-2). After 24 h, plasmid DNA was isolated, quantified and 0.5 ng of DNA plasmid from each condition was amplified with PCR (initial denaturation at 95 °C for 3 min, and then 95 °C for 15 s, 61 °C for 15 s and 72 °C for 15 s, and repeated for 33 cycles, and then a final extension at 72 °C for 5 min).

### 4.9. Radiation Treatment

Cells were irradiated, using a Linear Medical Accelerator ELEKTA SYNERGY, with dose at the isocenter of the beam of 2, 4, 6 and 8 Gy, Dose Rate 400 cGy/min. Dose was calculated by the “Masterplan Treatment Planning System”: 105 U.M. (Monitor Unit) for each 1 Gy provided. A Collapsed Cone Convolution algorithm was used, with photon energies of 6MV with a rectangular field of 14 × 30 cm^2^, with the technique of 2 opposite fields with gantry 90° and 270°, in order to have a homogeneous dose distribution. In addition, to improve dose homogeneity, cells were irradiated in a water-equivalent phantom to create full-scattered radiation condition in the field. Dose was verified with a Farmer ionization chamber. After radiation, cells were plated for colony-forming assay: for D341, 10,000 cells were plated in 6-well plates, to obtain around 500 colonies in control cells. For D283, 6000 cells were plated, while for ONS-76 and DAOY, 500 cells were plated, to obtain 300 colonies in control cells.

### 4.10. Cisplatin Treatment

Then, 10,000 D341 cells transfected with siCtrl or siCITK for 72 h, were treated with cisplatin at 1 and 10 μM concentration for 1 h, centrifugated at 150 g for 5 min and then plated for colony forming assay. For D283, 6000 cells were plated and transfected with siCtrl or siCITK for 72 h, and then cisplatin was added at 1 and 10 μM concentration for 1 h. After treatment, cells were washed 2 times with PBS, and fresh medium was added to allow colony growth. For ONS-76 and DAOY, 500 cells were plated in doxycycline containing medium, for 48 h, and then the cells were treated with cisplatin, similarly to D283 cell line.

### 4.11. Colony-Forming Assay

D341 clonogenic assay was stopped after 14 days. Since D341 colonies grow in suspension, Low Melting agarose was added, to obtain a final concentration of 0.7%. After agarose solidification, 1 mL of Nissl staining (0.1% Cresyl Violet Acetate and 0.6% glacial acetic acid) was added on top, and it was allowed to penetrate the gel and stain the colonies for 1–2 h and then de-stained, using H_2_O rinses. 

For D283, the clonogenic assay was stopped after 10 days, while for ONS-76 and DAOY, it was stopped after 5 days. After medium removal, colonies were stained for 10 min with Nissl staining and then rinsed in water. The surviving fraction was determined by counting colonies of more than 10 cells. 

In all experiments, the different conditions (e.g., radiation dose and cisplatin treatment) were compared to the correspondent siCtrl or siCITK without treatment.

### 4.12. Statistical Analysis

Statistical analyses were performed by using Microsoft Office Excel and GraphPad (Version 6, GraphPad Software, San Diego, CA, USA). Data are shown as the mean values of at least 3 independent experiments and standard error of the mean (mean ± SEM). Survival fraction data were fitted into a nonlinear regression curve, and siCtrl and siCITK curves were compared with Extra sum-of-squares F test. Mann–Whitney test was used to analyze γH2AX, 53BP1 and RAD51 foci. In all other cases, Unpaired two-tails Student’s *t*-test was used.

## 5. Conclusions

The most common pediatric brain tumor is medulloblastoma, which represents an important unmet medical challenge. In this study, we proposed CITK as a potential target for Group 3 and Group 4 medulloblastoma, since its downregulation induces cytokinesis failure, DNA-damage accumulation and apoptosis in cell models of these subgroups. Moreover, when combined with radiation or cisplatin, CITK knockdown is able to increase the effectiveness of these treatments. These effects were associated with reduced nuclear levels of RAD51 and an impairment of homologous recombination.

## Figures and Tables

**Figure 1 cancers-12-00542-f001:**
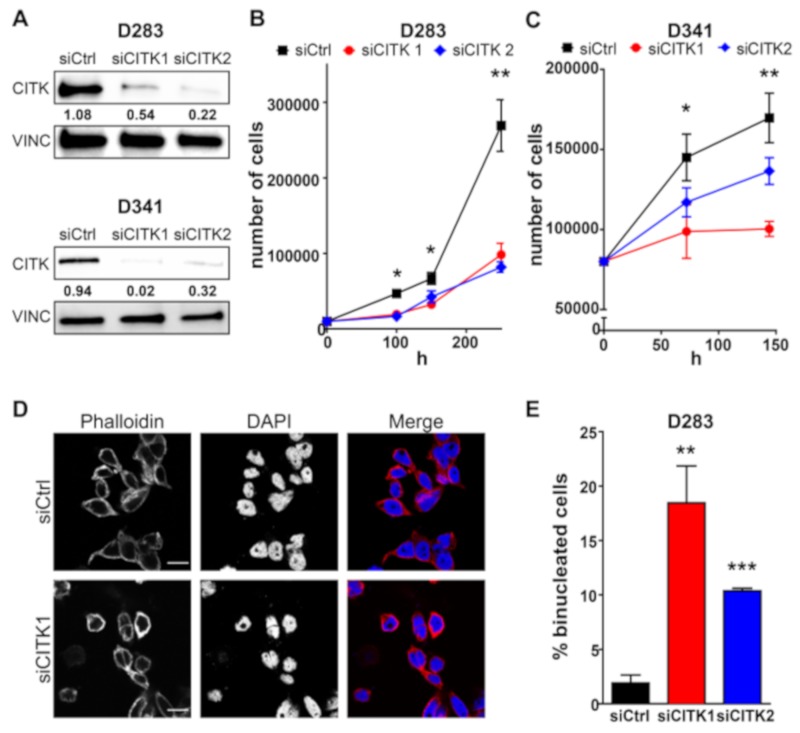
CITK knockdown decreases proliferation and induces cytokinesis failure in medulloblastoma cell lines. (**A**) Western blot analysis of total lysate from D283 and D341 cells, 72 h after treatment, with nontargeting (siCtrl) or CITK-specific siRNAs (siCITK1 and siCITK2). The level of CITK was analyzed, and the internal loading control was vinculin (VINC). The numbers below each band express the ratio between its normalized intensity and the average normalized intensity of the siCtrl. (**B**) 10,000 D283 cells were transfected with siCtrl, siCITK1 and siCITK2 and plated in triplicates. Growth curves were obtained by assessing cell number in each well, at 100, 150 and 200 h after transfection. (**C**) 80,000 D341 cells were transfected with siCtrl, siCITK1 and siCITK2 and plated in triplicates. Growth curves were obtained by assessing cell number in each well at 72 and 144 h after transfection. (**D**) Representative image of D283 cells processed for immunofluorescence 100 h after transfection with nontargeting or CITK-specific siRNA and stained with DAPI and anti-Phalloidin antibody. (**E**) Quantification of binucleated D283 cells in the indicated transfections, performed as in (**D**). (**F**) Representative image of D341 processed for immunofluorescence 72 h after transfection with nontargeting or CITK-specific siRNA and stained with DAPI and anti-Phalloidin antibody. (**G**) Quantification of binucleated D341 cells in the indicated transfections, performed as in (**F**). All quantifications were based on three independent biological replicates. Error bars, SEM. *, *p* < 0.05; **, *p* < 0.01, ***, *p* < 0.001; two-tailed Student’s *t*-test. Scale bars, 10 µm.

**Figure 2 cancers-12-00542-f002:**
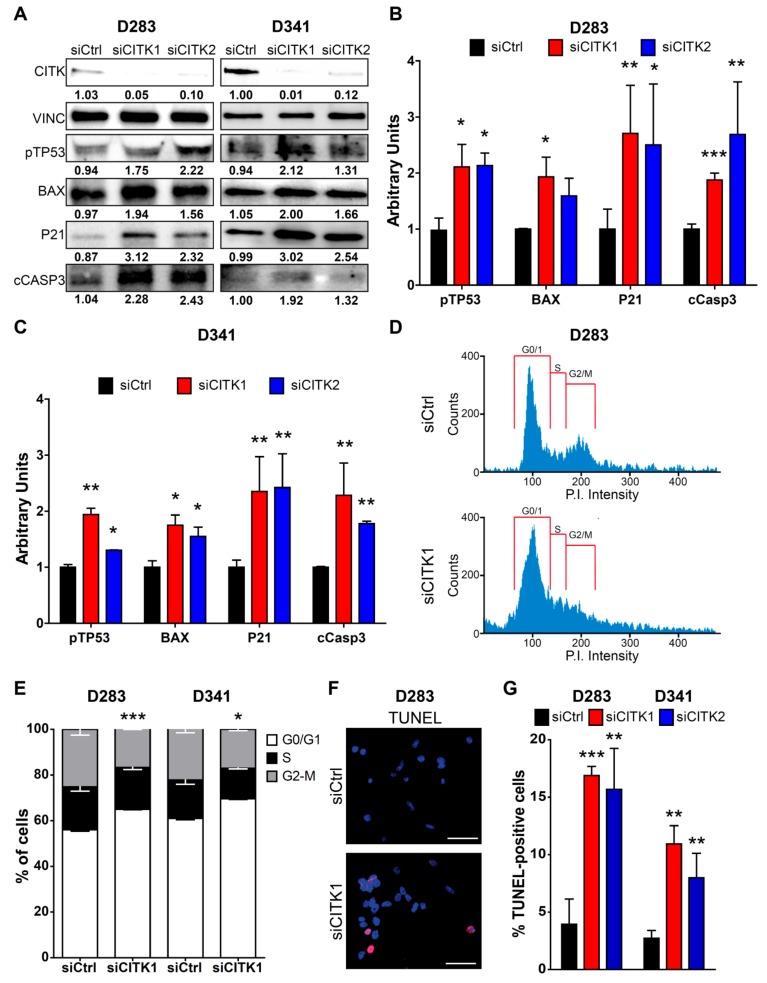
CITK knockdown leads to apoptosis and cell-cycle arrest in D283 and D341 cell lines. (**A**) Western blot analysis of total lysates from D283 and D341 cells, 100 and 72 h, respectively, after treatment with nontargeting (siCtrl) or CITK-specific siRNAs (siCITK1 and siCITK2). The level of CITK, pTP53, BAX, P21 and cCASP3 were analyzed, and the internal loading control was vinculin (VINC). (**B**,**C**) Quantification of the relative density of pTP53, BAX, P21 and cCASP3 in D283 and D341-treated cells, normalized to VINC and siCtrl average values. (**D**) Example of cell-cycle profile of D283 cells transfected with siCtrl and siCITK1 for 100 h. G0/G1, S and G2-M phases are shown. (**E**) Quantification of the percentage of D283 and D341 treated cells in the different cell-cycle phases. (**F**) Representative image of D283 processed for immunofluorescence 100 h after transfection with nontargeting or CITK-specific siRNA and stained with DAPI and TUNEL assay. (**G**) Quantification of the percentage of TUNEL positive nuclei in D283 and D341 treated cells. All quantifications were based on three independent biological replicates. Error bars, SEM. *, *p* < 0.05; **, *p* < 0.01, ***, *p* < 0.001; two-tailed Student’s *t*-test. Scale bars, 10 µm.

**Figure 3 cancers-12-00542-f003:**
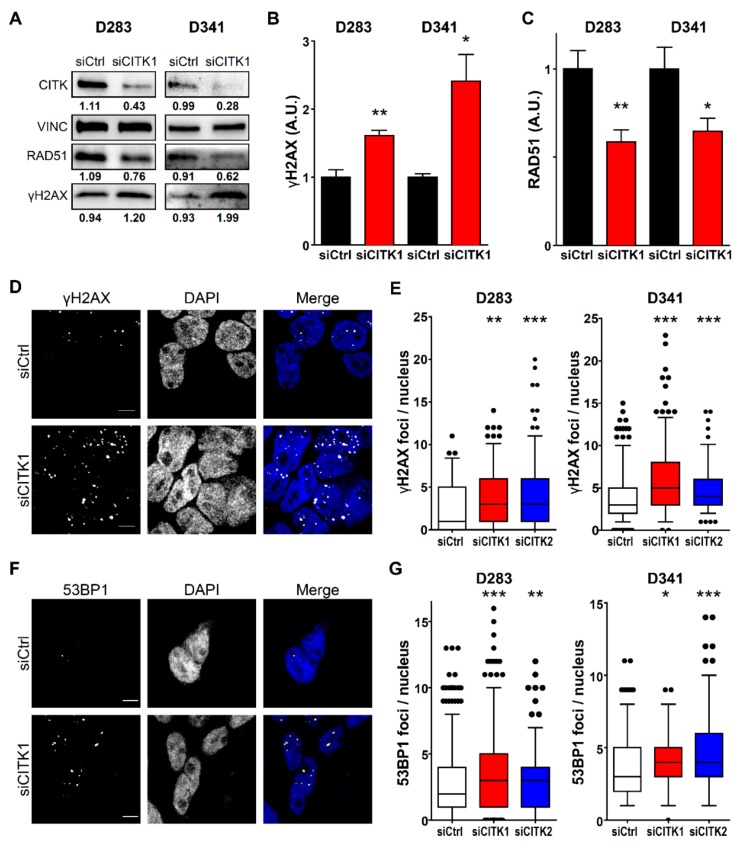
CITK knockdown induces RAD51 reduction and DNA-damage accumulation in G3/G4 MB cells. (**A**) Western blot analysis of total lysate from D283 and D341 cell lines, 100 and 72 h after treatment with siCtrl or siCITK1. The levels of CITK, RAD51 and γH2AX were analyzed. The internal loading control was vinculin (VINC). (**B**,**C**) Quantification of the relative density of γH2AX and RAD51 in D283 and D341 after CITK knockdown. (**D**,**F**) Representative images of D283 cells stained with DAPI and anti-γH2AX (**D**) or anti-53BP1 (**F**) antibodies 100 h after transfection with nontargeting or siCITK1. (**E**,**G**) Quantification of γH2AX (**E**) and 53BP1 (**G**) nuclear foci in D283 and D341 cells, treated as above. All quantifications were based on at least five independent biological replicates. Error bars, SEM. *, *p* < 0.05; **, *p* < 0.01; two-tailed Student’s *t*-test for blots. *, *p* < 0.05; **, *p* < 0.01, ***, *p* < 0.001 Mann–Whitney U test for γH2AX and 53BP1 foci. Scale bars, 5 µm. A.U., arbitrary unit.

**Figure 4 cancers-12-00542-f004:**
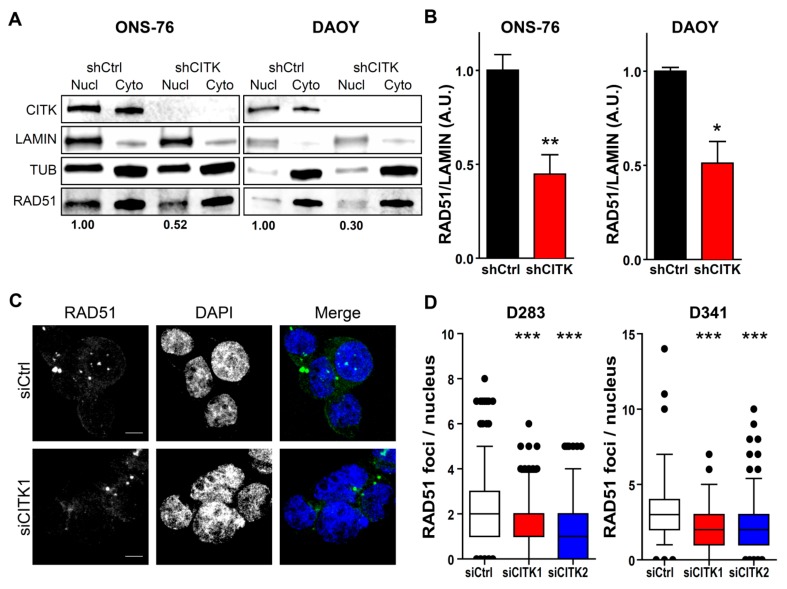
CITK knockdown reduces nuclear RAD51 and impairs homologous recombination. (**A**) Western blot analysis of nuclear (Nucl) and cytoplasmic (Cyto) fractions of ONS-76 and DAOY cells, expressing nontargeting sequence (shCtrl) or CITK-specific shRNA sequences under doxycycline-inducible control. Cells were analyzed 48 h after shRNAs induction with doxycycline-containing medium (2 μmol/L). The levels of CITK and RAD51 were analyzed. The internal loading control was Lamin A (LAMIN) for the nucleus and Tubulin (TUB) for cytoplasm. (**B**) Quantification of the relative density of RAD51 in ONS-76 and DAOY nuclei, normalized on Lamin A and average shCtrl levels. (**C**) Representative images of D283 cells stained with DAPI and anti-RAD51 antibody 72 h after transfection with nontargeting or CITK-specific siRNA. (**D**) Quantification of RAD51 foci in nuclei of D283 and D341 cells treated with the indicated siRNAs. (**E**) Semiquantitative analysis of homologous recombination products generated in CITK-knockdown D283 and D341 cells, 100 and 72 h after transfection with the indicated siRNAs, along with recombinogenic dl-1 and dl-2 plasmids. A PCR for the total dl1 and dl2 sequences was performed as internal control of transfection efficiency. (**F**) Quantification of the homologous recombination product formation in D283, D341, ONS-76 and DAOY treated cells, normalized on the internal controls. All quantifications were based on at least three independent biological replicates. Error bars, SEM. *, *p* < 0.05; **, *p* < 0.01, ***, *p* < 0.001; two-tailed Student’s *t*-test. ***, *p* < 0.001 Mann–Whitney U test for RAD51 foci. Scale bars, 5 µm.

**Figure 5 cancers-12-00542-f005:**
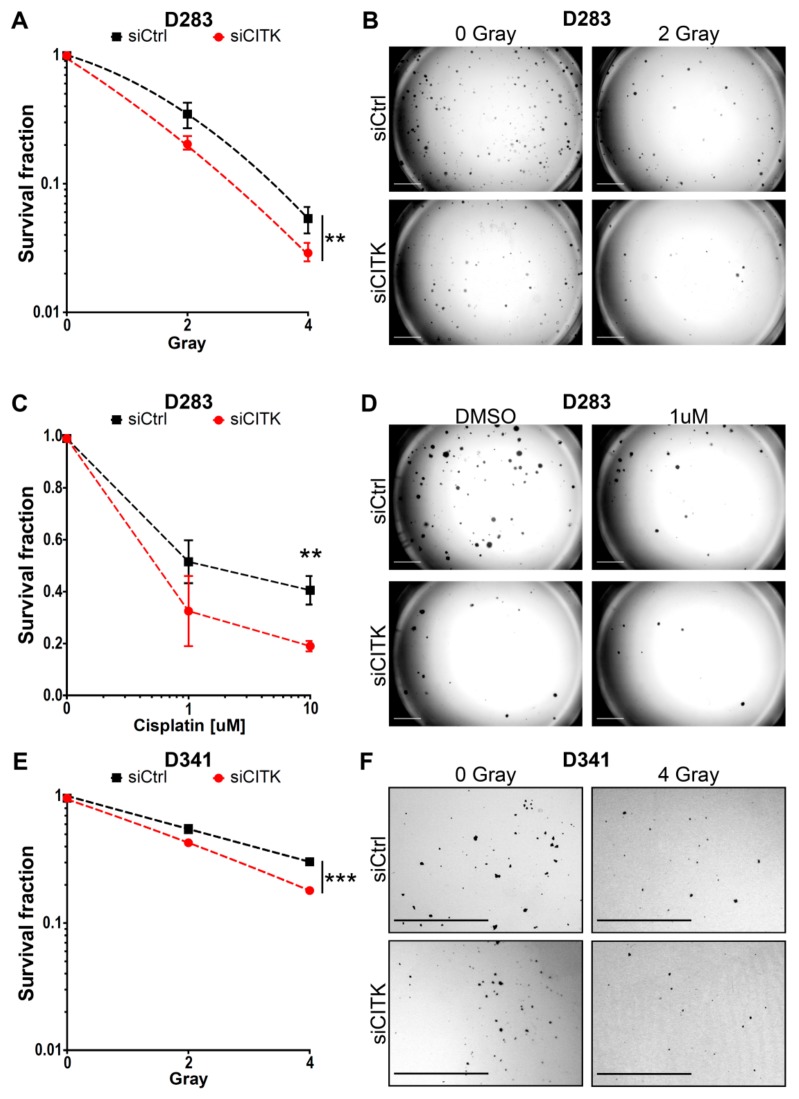
CITK knockdown potentiates the effects of radiation and cisplatin treatment in G3/G4 MB cell lines. (**A**) D283 cells were transfected with nontargeting or CITK-specific siRNA and, after 72 h, irradiated at the indicated doses and plated at low density (see methods for the details). After 10 days, the colonies were fixed and stained with crystal violet. The siCtrl and siCITK curves were obtained by fitting the values into a nonlinear regression curve and compared with Extra sum-of-squares F test. (**B**) Phase contrast images of representative fields of D283 transfected with siCtrl or siCITK 10 days after irradiation. (**C**) D283 cells were plated and transfected with nontargeting or CITK-specific siRNA. After 72 h, cells were treated with 1 or 10 µM of cisplatin for 1 h. Cells were then washed, and fresh medium was added. Colonies were stained after 10 days with crystal violet. (**D**) Phase contrast images of representative fields of D283 transfected with siCtrl or siCITK, 10 days after treatment with 1 µM of cisplatin. (**E**) D341 cells were transfected with nontargeting or CITK-specific siRNA and after 72 h irradiated at the indicated doses. After 14 days, the colonies were fixed and stained with crystal violet. The curves were obtained and analyzed as in (**A**). (**F**) Phase contrast images of representative fields of D341 transfected with siCtrl or siCITK 14 days after irradiation. (**G**) D341 cells were plated and transfected with nontargeting or CITK-specific siRNA. After 72 h, cells were treated with 1 or 10 µM of cisplatin for 1 hour. Cells were then washed, and fresh medium was added. Colonies were stained after 14 days, with crystal violet. (**H**) Phase contrast images of representative fields of D341 transfected with siCtrl or siCITK 14 days after treatment with 1 µM of cisplatin. All quantifications were based on five independent biological replicates. Error bars, SEM. *, *p* < 0.05; **, *p* < 0.01; ***, *p* < 0.001, two-tailed Student’s *t*-test.

**Figure 6 cancers-12-00542-f006:**
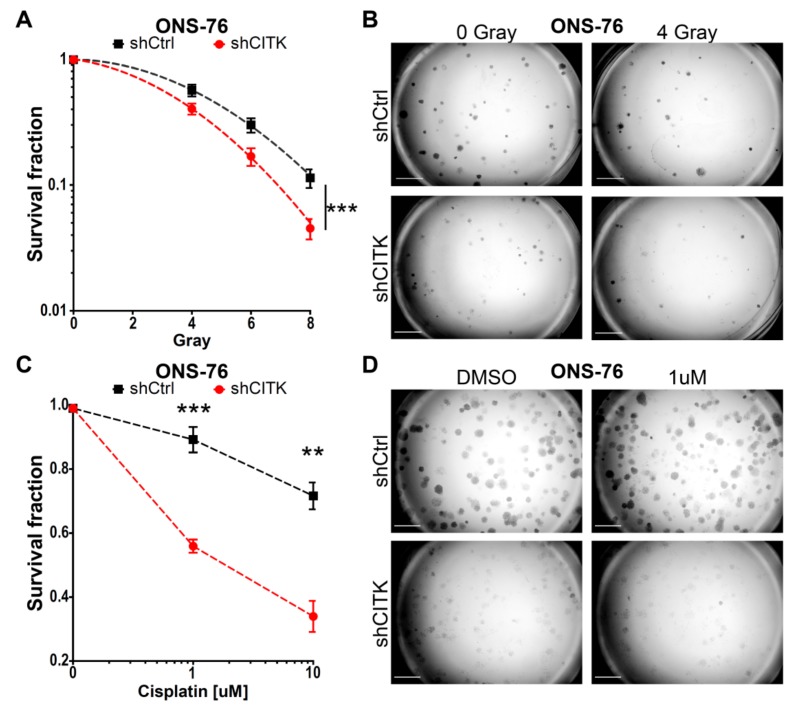
CITK knockdown potentiates the effects of radiation and cisplatin treatment in ONS-76 and DAOY cell lines. (**A**) ONS-76 stably transfected with inducible nontargeting (shCtrl) or CITK-specific shRNA sequences were plated in doxycycline-containing medium (2 μmol/L) and irradiated at the indicated doses 48 h later. After five days, colonies were fixed and stained with crystal violet. The shCtrl and shCITK curves were obtained by fitting the values into a nonlinear regression model, and curves were compared with Extra sum-of-squares F test. (**B**) Phase contrast images of representative fields of ONS-76 expressing shCtrl and shCITK five days after radiation. (**C**) ONS-76 cells expressing shCtrl or CITK-specific shRNA were plated in doxycycline-containing medium (2 μmol/L) and, after 48 h, treated with 1 or 10 μM of cisplatin for 1 hour. Cells were then washed, and fresh medium was added. Colonies were stained after five days, with crystal violet. (**D**) Phase contrast images of representative fields obtained from ONS-76 cells expressing shCtrl and shCITK five days after 1 μM of cisplatin treatment. (**E**) DAOY stably transfected with inducible nontargeting (shCtrl) or CITK-specific shRNA were plated in doxycycline-containing medium (2 μmol/L) and, after 48 h, irradiated at the indicated doses. After five days, the colonies were fixed and stained with crystal violet. The curves were obtained and analyzed as in (**A**). (**F**) Phase contrast images of representative fields of DAOY expressing shCtrl and shCITK five days after radiation. (**G**) DAOY cells expressing shCtrl or CITK-specific shRNA were plated in doxycycline-containing medium (2 μmol/L) and, after 48 h, treated with 1 or 10 μM of cisplatin for 1 h. Cells were then washed, and fresh medium was added. Colonies were stained after five days, with crystal violet. (**H**) Phase contrast images of representative fields of DAOY expressing shCtrl and shCITK five days after 1 uM of cisplatin treatment. All quantifications were based on five independent biological replicates. Error bars, SEM. **, *p* < 0.01; ***, *p* < 0.001; two-tailed Student’s *t*-test for each point.

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
