# Peer review of "CITK Loss Inhibits Growth of Group 3 and Group 4 Medulloblastoma Cells and Sensitizes Them to DNA-Damaging Agents"

_cancers, 2020, doi:10.3390/cancers12030542_

Round 1

Reviewer 1 Report

The authors have answered satisfactorily to most of the previous points that were raised. However, two concerns remain:

1/The synergistic effect is not demonstrated. Synergism is defined when a combination of two treatments have more effect that the addition of each, individually. As it is presented (effect of sictl and siCITK set at 1), it is not possible to draw this conclusion, although it is not questionable that the combination is more potent that each treatment individually. I suggest to change the term and not use synergism.

2/Since HR is effective mainly in S/G2 phase of the cell cycle and that the siCITKs decrease the number of cells in these phases, one may wonder if the effect on HR is not a direct consequence of the G0/G1 arrest. To demonstrate the direct effect on HR and to exclude this possibility, it would be important to demonstrate that Cdk4/6 inhibitors, for example or other blockers in G0/G1, do not affect DSB nor decrease HR to the same level.

Minor points:

- lane 133 phospho-TP53 is misspelled

-Lane 242, "In these plots" incorrect fig5 and 6 are not plots.

Author Response

The authors have answered satisfactorily to most of the previous points that were raised.

We thank the Referee for the overall positive judgement of our revised manuscript.

However, two concerns remain:

1/The synergistic effect is not demonstrated. Synergism is defined when a combination of two treatments have more effect that the addition of each, individually. As it is presented (effect of sictl and siCITK set at 1), it is not possible to draw this conclusion, although it is not questionable that the combination is more potent that each treatment individually. I suggest to change the term and not use synergism.

We do not totally agree with the Referee because, in our opinion, our data show that the combined effect is more than additive. However, we recognize that the differences are not strong enough to make a big point about synergism. For this reason, in the amended text we carefully avoided any term related to synergism, using instead words such as “strongly”, “potentiated” and “combined”.

2/Since HR is effective mainly in S/G2 phase of the cell cycle and that the siCITKs decrease the number of cells in these phases, one may wonder if the effect on HR is not a direct consequence of the G0/G1 arrest. To demonstrate the direct effect on HR and to exclude this possibility, it would be important to demonstrate that Cdk4/6 inhibitors, for example or other blockers in G0/G1, do not affect DSB nor decrease HR to the same level.

We think that this request goes beyond the scope of the present work. A careful experimental assessment of this point (to which we are very interested) would require much deeper mechanistic investigation. Indeed, even the experiment suggested by the Referee, would only provide indirect evidence. Moreover, we are not sure that Cdk inhibitors are completely devoid of direct effects on HR machinery. Nevertheless, we agree that the point raised by the Referee must be underscored. To this aim, we have added to the discussion the following statements (Pag. 17):

“Moreover, considering the cell cycle block which we detected in CITK knockdown cells, it is also possible that the reduced levels of RAD51 and HR efficiency are a consequence of DSB-dependent cell cycle arrest, rather than a cause DSB accumulation. Indeed, it is well known that RAD51 expression and HR efficiency are highest in cycling cells (PMID: 9697414). Although more studies are clearly necessary to unravel the mechanistic relationships between these phenomena, our results further underscore the central role of DNA damage and repair in the biological role of CITK in dividing cells.”

Minor points:

- lane 133 phospho-TP53 is misspelled

-Lane 242, "In these plots" incorrect fig5 and 6 are not plots.

We thank the Referee for careful reading, the errors have been amended.

Reviewer 2

In the second version of the manuscript, the authors put efforts in answering my previous concerns, which are revised in an adequate manner. Furthermore, additional experiments performed clearly improved the significance of the manuscript.

We thank the Referee for the positive judgement of our revised manuscript.

Reviewer 2 Report

In the second version of the manuscript, the authors put efforts in answering my previous concerns, which are revised in an adequate manner. Furthermore, additional experiments performed clearly improved the significance of the manuscript.

Author Response

In the second version of the manuscript, the authors put efforts in answering my previous concerns, which are revised in an adequate manner. Furthermore, additional experiments performed clearly improved the significance of the manuscript.

We thank the Referee for the positive judgement of our revised manuscript.

This manuscript is a resubmission of an earlier submission. The following is a list of the peer review reports and author responses from that submission.

Round 1

Reviewer 1 Report

Pallavicini et al. investigate the involvement of the citron kinase (CITK) in group3 (G3) and group 4 (G4) medulloblastoma (MB). This work is an extension of a previous work from the same group showing that citron kinase inactivation slows down SHH-MB progression by inducing DNA damage. The present work extends this notion to G3 and G4 MB and shows a potential radiosensitization effect. The novelty of this work is minor and the question of the citron kinase as a potential therapeutic target remains unclear since it is required for normal neural progenitors and a potential therapeutic window unassessed. The synergistic effect of CITK depletion with DNA damaging agents is not demonstrated nor the involvement of the decrease of RAD51.

Major points:

1/ Although this siRNA has been previously published, two independent siRNA should be used throughout the study or rescue experiments performed to exclude potential off target effects.

2/ Growth curves are performed at very late time points after transfection, especially for the D283 cell line. One may question if the siRNA is still effective at this time point. Seeding the cell lines at higher concentration may overcome this issue. Moreover, cell cycle profile should be performed to assess whether CITK depletion also alters cell cycle progression.

3/ To sustain IF data, FACS analysis should be performed to assess DNA content and demonstrate by another mean the presence of binucleated cells (number of cells ≥4n).

4/ For reliable quantification, TP53 levels and cleaved-casp3 should also be assessed by WB analysis. Expression of some downstream p53 targets (BAX, MDM2, PUMA, NOXA, p21…) should be investigated by RT-qPCR to demonstrate that p53 is indeed active. The level of p21 by IF is not sufficient.

5/ The number of γH2AX and RAD51 foci should be quantified by IF. The overall level of these proteins is not completely informative.

6/ Figure 4, more convincing blots should be shown. Indeed, they do not faithfully represent the quantification. Regarding the quantification, we ignore if the ratio of nuclear RAD51/lamin, as it should, is presented. Moreover, since the paper is centered on G3/G4 MB, this experiment should be performed on D341 and D283 cell lines.

7/The synergistic effect of CITK depletion with DNA damaging agents is not demonstrated.

8/ No statistical information is provided on Fig5A-5E and 6A-6E. Is there any statistical difference between shCtl vs shCITK upon irradiation? Consequently, the synergistic (or even additive) effect of CITK with irradiation is unclear.

9/ It is proposed that CITK depletion sensitizes MB cells by reducing RAD51 level but not demonstrated. Rescue experiments by reintroducing RAD51 or, at least, RAD51 KD demonstrating that it mimics CITK depletion should be shown. 

Minor points:

1/The attribution of the different cell lines to specific group can be questioned. Although ONS76 and DAOY likely originate from a SHH-MB, they do not depend anymore on the SHH signaling pathway and are not considered as good model for this group of MB. D283Med have been classified as a G4 but also as G3 depending on the study. I suggest to name G3/G4 vs non G3/G4 models.

2/ in Lane 48-49 it is mentioned "… but they are usually around 50% in most other cases, with worse prognosis in Group 3 and Group 4 patients [3]". This statement is incorrect. G4 displays an intermediate prognosis.

Author Response

Reviewer 1

Pallavicini et al. investigate the involvement of the citron kinase (CITK) in group3 (G3) and group 4 (G4) medulloblastoma (MB). This work is an extension of a previous work from the same group showing that citron kinase inactivation slows down SHH-MB progression by inducing DNA damage. The present work extends this notion to G3 and G4 MB and shows a potential radiosensitization effect. The novelty of this work is minor and the question of the citron kinase as a potential therapeutic target remains unclear since it is required for normal neural progenitors and a potential therapeutic window unassessed. The synergistic effect of CITK depletion with DNA damaging agents is not demonstrated nor the involvement of the decrease of RAD51.

We thank the Referee for the careful evaluation of the manuscript and for the very constructive criticisms.

Major points:

1/ Although this siRNA has been previously published, two independent siRNA should be used throughout the study or rescue experiments performed to exclude potential off target effects.

We have repeated most of the crucial experiments using a second, previously validated sequence (siCITK2). In particular, we assessed its effects on cell proliferation and apoptosis, in D283 and D341 cell lines. We have also used this sequence, in parallel with siCITK1, in most of the new experiments. As expected from our previous work, the results which we obtained with the two sequences are very similar, although siCITK2 is generally not as effective as siCITK1.

2/ Growth curves are performed at very late time points after transfection, especially for the D283 cell line. One may question if the siRNA is still effective at this time point. Seeding the cell lines at higher concentration may overcome this issue…..

D283 and D341 cells have population doubling times of 53 and 37 hours, respectively. For this reason, it is difficult to highlight significant differences at early time points, even at higher cell concentrations. Nevertheless, as shown in Figure S1, WB analysis revealed that CITK expression is still downregulated at long times after transfection, i.e. 200 hours in D283 cells and 144 hours in D341 cells.

….Moreover, cell cycle profile should be performed to assess whether CITK depletion also alters cell cycle progression. 3/ To sustain IF data, FACS analysis should be performed to assess DNA content and demonstrate by another mean the presence of binucleated cells (number of cells).

As suggested by the Referee, we have analyzed by flow cytometry the effect of CITK knockdown on cell cycle, in D283 and D341 cells. In both lines, the strongest effect which we detected was a significant increase of the G0/G1 fraction (Fig. 3D-E). This cell cycle arrest is consistent with our previous observations in DAOY and ONS-76 cells (Pallavicini et al. 2018). In contrast, the fraction of cells with DNA content ≥4n was not increased. To provide more support to the possibility that binucleated cells are the result of cytokinesis failure, rather than aberrations of nuclear morphology, we measured total DAPI intensity at the single cell level. As expected, this analysis revealed that, on average, the DAPI intensity in binucleated cells was double than in mononucleated cells (Fig. S1B). We think that the two results are not in contrast, but provide complementary information on the two main effects of CITK depletion: cell cycle arrest and cytokinesis failure. Most likely, FACS analysis is unable to highlight the increased frequency of binucleation because less cells are capable to progress into mitosis. Under this scenario, the tetraploid peak detected by flow cytometry is probably overestimating the fraction of cells that are still actively dividing.

4/ For reliable quantification, TP53 levels and cleaved-casp3 should also be assessed by WB analysis. Expression of some downstream p53 targets (BAX, MDM2, PUMA, NOXA, p21…) should be investigated by RT-qPCR to demonstrate that p53 is indeed active. The level of p21 by IF is not sufficient.

As suggested by the Referee, we have analyzed by western blot the levels of phospho-TP53, of the TP53 targets BAX and P21 and of cCasp3 (Fig. 2A-B), after transfection of both siCITK1 and siCITK2. The obtained results support the conclusion that CITK knockdown leads to robust activation of the p53 pathway and increased apoptosis.

5/ The number of γH2AX and RAD51 foci should be quantified by IF. The overall level of these proteins is not completely informative.

As suggested by the Referee, we have analyzed γH2AX and RAD51 foci after transfection of both siCITK1 and siCITK2. The results are shown in Fig. 3D-G.

6/ Figure 4, more convincing blots should be shown. Indeed, they do not faithfully represent the quantification. Regarding the quantification, we ignore if the ratio of nuclear RAD51/lamin, as it should, is presented. Moreover, since the paper is centered on G3/G4 MB, this experiment should be performed on D341 and D283 cell lines.

We confirm that quantifications were (and are) normalized on lamin levels. This is now clearly stated in the figure legend. Moreover, we replaced the RAD51 blot with a more representative exposure. Concerning RAD51 nuclear levels, we agree with the Referee. In consideration of the technical difficulties of performing the biochemical measurement, we measured nuclear RAD51 foci in D341 and D283 cell lines by IF analysis (Fig. 4C-D).

7/The synergistic effect of CITK depletion with DNA damaging agents is not demonstrated.

To highlight the possible synergistic effects of CITK knockdown with IR or Cisplatin treatments, we compared the reduction of colony-forming efficiency induced by treatments in absence or presence of anti-CITK siRNA (Fig. 5 and 6). In these plots, cells transfected with control and anti-CITK siRNAs without treatments were set as reference for the same transfections plus IR or Cisplatin treatments. Therefore, the strength of the synergistic effect is represented by the increased slope of the combined treatment curve (siCITK and DNA damaging agents), as compared to the single treatment curve (siCtrl and DNA damaging agents). We apologize for not making the point more clearly in the previous version, and hope that the present explanation (which has been integrated in the text) is more satisfactory.

8/ No statistical information is provided on Fig5A-5E and 6A-6E. Is there any statistical difference between shCtl vs shCITK upon irradiation? Consequently, the synergistic (or even additive) effect of CITK with irradiation is unclear.

We generated plots by fitting the values into a nonlinear regression curve and compared them with Extra sum-of-squares F test. The statistically significant differences are now also showed in the graphs.

9/ It is proposed that CITK depletion sensitizes MB cells by reducing RAD51 level but not demonstrated. Rescue experiments by reintroducing RAD51 or, at least, RAD51 KD demonstrating that it mimics CITK depletion should be shown.

We agree with the Referee about the importance of this point and we tried to perform rescue experiments using a RAD51 expression plasmid. However, we faced the technical challenge that plasmid transfection, by itself, produced an increase in the background level of foci much higher than the effect produced by CITK knockdown. On the other hand, we think that also the KD experiment would be not decisive, because it has been widely established that RAD51 KD leads to accumulation of DSB (Liu et al., 2019; Russell et al., 2003; Zhang et al., 2018; Zhong et al., 2016). To overcome these difficulties, in agreement with the suggestion of the second Referee, we thought to obtain direct functional evidence that CITK knockdown is detrimental for the HR activity. This turned out to be the case, with all lines (Fig. 4D-E). We realize that the mechanistic link with RAD51 is not formally proved, but we hope that the demonstration of reduced HR activity could make the point less essential for supporting our general conclusions.

Minor points:

1/The attribution of the different cell lines to specific group can be questioned. Although ONS76 and DAOY likely originate from a SHH-MB, they do not depend anymore on the SHH signaling pathway and are not considered as good model for this group of MB. D283Med have been classified as a G4 but also as G3 depending on the study. I suggest to name G3/G4 vs non G3/G4 models.

We agree with referee and we changed the text accordingly

2/ in Lane 48-49 it is mentioned "… but they are usually around 50% in most other cases, with worse prognosis in Group 3 and Group 4 patients [3]". This statement is incorrect. G4 displays an intermediate prognosis.

We thank the referee for the advice; we changed the text accordingly.

Reviewer 2 Report

The purpose of the manuscript presented by Pallavicini et al. covers the investigation of the impact of Citron kinase (CITK) as a molecular target in Group 3 and Group 4 medulloblastoma (MB) cancer cells. By using RNA interference in D341 and D283 lines, authors report  on an impairment of proliferation, cytokinesis failure, accumulation of DNA damage, reduced RAD51 levels and apoptosis induction in CITK knockdown (KD) cells. Moreover, additional treatment with irradiation and cisplatin resulted in a reduced clonogenic survival of all MB lines tested.

Although the topic on a Group 3 and Group 4 MB specific molecular targeting cover an up-to date and interesting area of research as compared to preceding  publications of the group, however, the manuscript suffers from a multitude of significant shortcomings and weak performance that require

Major points of criticism:

Figure 2 B/D: Percentages of TP53 and cCASP3 levels could not be traced from the examples given in Figure 2B. Moreover, apoptosis induction has to be confirmed by at least a second method, like AnnexinV/PI staining or TUNEL assay as has been done in a recent manuscript of the group (Cancer Res 2018). Authors indicate CITK KD to impair cytokinesis but failed to provide data on the cell cycle distribution. Authors used a densitometric assessment of yH2AX and RAD51 detection, however, the more appropriate measurement is quantification of nuclear yH2AX and RAD51 foci as has been done for 53BP1 staining in Figure 3G. Figure 5A/C. The radiation sensitizing effect of CITK is not convincing and apparently does not reach a level of significance as given in the graphs. It is not tractable to the reviewer why authors restricted their analyses to a 2 and 4 Gy irradiation in figure 5A. In addition, plating efficiencies should be given. Authors restricted their analysis on the combined modality treatment to a simple clonogenic survival assay. From a translational point of view, however it is of importance whether or not CITK KD impacts on DNA-repair by HR, cell cycle distribution and apoptosis induction. In line with that, authors should indicate by HR activity assay that this repair pathway indeed is impaired upon CITK KD and radiation treatment. Discussion section: Authors extensively discussed the interrelationship of CITK and RAD51/HR repair pathway. To substantiate their findings, perfoming HR activity assays (e.g. Crispr-Cas9 HR Reporter system) is crucial.

Author Response

Reviewer 2

The purpose of the manuscript presented by Pallavicini et al. covers the investigation of the impact of Citron kinase (CITK) as a molecular target in Group 3 and Group 4 medulloblastoma (MB) cancer cells. By using RNA interference in D341 and D283 lines, authors report  on an impairment of proliferation, cytokinesis failure, accumulation of DNA damage, reduced RAD51 levels and apoptosis induction in CITK knockdown (KD) cells. Moreover, additional treatment with irradiation and cisplatin resulted in a reduced clonogenic survival of all MB lines tested.

Although the topic on a Group 3 and Group 4 MB specific molecular targeting cover an up-to date and interesting area of research as compared to preceding  publications of the group, however, the manuscript suffers from a multitude of significant shortcomings and weak performance that require

We thank the Referee for the careful evaluation of the manuscript and for the very constructive criticisms.

Major points of criticism:

Figure 2 B/D: Percentages of TP53 and cCASP3 levels could not be traced from the examples given in Figure 2B. Moreover, apoptosis induction has to be confirmed by at least a second method, like AnnexinV/PI staining or TUNEL assay as has been done in a recent manuscript of the group (Cancer Res 2018).

To strengthen the results, we have analyzed by western blot the levels of phospho-TP53, of the TP53 targets BAX and P21 and of cCasp3 (Fig. 2A-B), after transfection of both siCITK1 and siCITK2. Moreover, as suggested by the Referee, we have measured apoptosis induction also through TUNEL assay (Fig. 3F-G).

Authors indicate CITK KD to impair cytokinesis but failed to provide data on the cell cycle distribution.

As suggested by the Referee, we have analyzed by flow cytometry the effect of CITK knockdown on cell cycle, in D283 and D341 cells. In both lines, the strongest effect which we detected was a significant increase of the G0/G1 fraction (Fig. 3D-E). This cell cycle arrest is consistent with our previous observations in DAOY and ONS-76 cells (Pallavicini et al. 2018). In contrast, the fraction of cells with DNA content ≥4n was not increased. To provide more support to the possibility that binucleated cells are the result of cytokinesis failure, rather than aberrations of nuclear morphology, we measured total DAPI intensity at the single cell level. As expected, this analysis revealed that, on average, the DAPI intensity in binucleated cells was double than in mononucleated cells (Fig. S1B). We think that the two results are not in contrast, but provide complementary information on the two main effects of CITK depletion: cell cycle arrest and cytokinesis failure. Most likely, FACS analysis is unable to highlight the increased frequency of binucleation because less cells are capable to progress into mitosis. Under this scenario, the tetraploid peak detected by flow cytometry is probably overestimating the fraction of cells that are still actively dividing.

Authors used a densitometric assessment of yH2AX and RAD51 detection, however, the more appropriate measurement is quantification of nuclear yH2AX and RAD51 foci as has been done for 53BP1 staining in Figure 3G.

We agree with the Referee and measured nuclear RAD51 foci in D341 and D283 cell lines by IF analysis (Fig. 4C-D)

Figure 5A/C. The radiation sensitizing effect of CITK is not convincing and apparently does not reach a level of significance as given in the graphs. It is not tractable to the reviewer why authors restricted their analyses to a 2 and 4 Gy irradiation in figure 5A. In addition, plating efficiencies should be given. Authors restricted their analysis on the combined modality treatment to a simple clonogenic survival assay. From a translational point of view, however it is of importance whether or not CITK KD impacts on DNA-repair by HR, cell cycle distribution and apoptosis induction. In line with that, authors should indicate by HR activity assay that this repair pathway indeed is impaired upon CITK KD and radiation treatment.

To highlight the possible synergistic effects of CITK knockdown with IR or Cisplatin treatments, we compared the reduction of colony-forming efficiency induced by treatments in absence or presence of anti-CITK siRNA (Fig. 5 and 6). In these plots, cells transfected with control and anti-CITK siRNAs without treatments were set as reference for the same transfections plus IR or Cisplatin treatments. Therefore, the strength of the synergistic effect is represented by the increased slope of the combined treatment curve (siCITK and DNA damaging agents), as compared to the single treatment curve (siCtrl and DNA damaging agents). We generated curves by fitting the values into a nonlinear regression curve and compared them with Extra sum-of-squares F test. The statistically significant differences are now also showed in the graphs. We apologize for not making the point more clearly in the previous version, and hope that the present explanation (which has been integrated in the text) is more satisfactory.

We agree on the importance of assessing the effects of CITK KD on HR. To assess this point, we used a plasmid-based HR assay (NORGEN HR Kit (Lee et al., 2018; Misic et al., 2016; Ohba et al., 2014)). This test confirmed that HR is impaired by CITK loss, in all the tested MB lines (Fig. 4D-E).

Discussion section: Authors extensively discussed the interrelationship of CITK and RAD51/HR repair pathway. To substantiate their findings, perfoming HR activity assays (e.g. Crispr-Cas9 HR Reporter system) is crucial.

We think that our conclusions are now better justified, based on the new experiments reported in Fig. 4D-E.